# Weighted Information Models for the Quantitative Prediction and Evaluation of the Geothermal Anomaly Area in the Plateau: A Case Study of the Sichuan–Tibet Railway

**Wenbo Zhao** [1,2] **, Qing Dong** [1,*] **, Zhe Chen** [3,4] **, Tao Feng** [5] **, Dong Wang** [5] **, Liangwen Jiang** [5] **, Shihui Du** [6] **, Xiaoyu Zhang** [6] **, Deli Meng** [1,2] **, Min Bian** [1,2] **and Jianping Chen** [3,4]

1    Key Laboratory of Digital Earth Science, Aerospace Information Research Institute, Chinese Academy of Sciences, Beijing 100094, China; zhaowb@aircas.ac.cn (W.Z.); mengdl@aircas.ac.cn (D.M.); bianmin19@mails.ucas.ac.cn (M.B.)
2    University of Chinese Academy of Sciences, No.19(A) Yuquan Road, Shijingshan District, Beijing 100049, China
3    School of Earth Science and Resources, China University of Geosciences (Beijing), Beijing 100083, China; 2101180003@cugb.edu.cn (Z.C.); 3s@cugb.edu.cn (J.C.)
4    Beijing Key Laboratory of Land and Resources Information Research and Development, Beijing 100083, China
5    China Railway Eryuan Engineering Group Co. Ltd., Chengdu 610031, China; fengtao@ey.crec.cn (T.F.); wangdong13@ey.crec.cn (D.W.); jianglw@ey.crec.cn (L.J.)
6    China Railway First Survey and Design Institute Group Co. Ltd., Xi'an 710043, China; dldsh@fsdi.com.cn (S.D.); dlzxy@fsdi.com.cn (X.Z.)
*    Correspondence: dongqing@aircas.ac.cn; Tel.: +86-10-82178121

**Abstract:** The prediction of geothermal high-temperature anomalies along the plateau railway will be helpful in the construction of the project and its later management. Taking the Sichuan–Tibet railway as the study area and based on Landsat8 thermal infrared images, map data, and measured data regarding the cause and distribution of geothermal high-temperature anomalies, through correlation analysis, we selected six impact factors including the LST, combined entropy of geological formation, fault density, buffer distance to rivers, magnetic anomaly, and earthquake peak acceleration as the input maps of the model. The index-overlay information model, the weights of the entropy information model, and the weights of the evidence information model were established to quantitatively predict the geothermal anomaly in the study area, and the prediction maps were divided into four classes. The results show that the weights of the evidence information model achieved a high prediction accuracy; the success index and the ratio of the high anomaly area reached 0.0053% and 0.872, respectively, and the spatial distribution of the geothermal points is basically consistent with the prediction results. This research can act as a reference for the design and construction of the Sichuan–Tibet railway.

**Keywords:** quantitative prediction; information model; geothermal anomaly area; combined entropy of geological formation; weights of evidence

---

## 1. Introduction

High-temperature geothermal anomalies will seriously affect the health and safety of operators and the efficiency of mechanical equipment in the process of railway construction, reduce the durability of building materials, and threaten the stability and safety of deep excavation tunnels [1]. The Sichuan–Tibet railway is a major strategic project as part of China's implementation of a comprehensive transportation system for the development of the western region. The whole line starts from Chengdu in the east and passes Ya'an, Kangding, Changdu, Linzhi, Shannan, and Lhasa in the west, with a total length of about 1543 km. The railway must pass through an area with the most complex geological and geomorphic environments in the world. It has the characteristics of significant terrain

height differences, strong plate activity, frequent mountain disasters, and a sensitive ecological environment [2–7]. Moreover, due to the large length and buried depth of the railway tunnel, its construction and maintenance process is greatly affected by high-temperature disasters [8]. Therefore, it is very important to predict and evaluate the geothermal high-temperature anomaly area using modern space exploration technology and a spatial analysis method.

In the study of geothermal anomaly prediction, common methods include the thermal infrared remote sensing inversion method, the mathematical statistical model, the spatial information superposition analysis method, and the geophysical method [9–11]. These methods have some problems, such as their single-factor consideration, the complex field and indoor work required, their large proportion of human subjective factors, and their low recognition accuracy [12]. Therefore, it is very important to build a multifactor prediction model to improve the automation and delineation accuracy in identifying geothermal high-temperature anomaly areas. Data-driven or knowledge-driven models based on a Geographic Information System (GIS) and Remote Sensing (RS), such as multiple regression analysis, logistic regression analysis, the information method, and artificial neural networks, have been widely used in metallogenic prediction and evaluation [13], groundwater resource evaluation and prediction [14], landslide risk assessment [15], pollution risk assessment [16], and other fields. For example, Proll-Ledesma used faults, hot springs, resistivity, and other factors to build a Boolean, index-overlay, and fuzzy combination model to evaluate the Los Azufres geothermal area in Mexico; the results show that the highly geothermal area prediction is consistent with the existing geothermal point location [17]. Coolbaugh combined multiple factors to build a logistic regression model to predict and evaluate geothermal areas in Nevada and the western basin; 170 potential geothermal zones were found in Nevada [18]. Using seismic, active fault, Bouguer gravity anomaly, and magnetic anomaly data; a tectonic index-overlay model; and a weight of evidence model, Tüfekçi et al. evaluated the geothermal area of the Anatolia continent where Turkey is located [19]. Noorollahi et al. developed a model for geothermal resource exploration by using fault, hot spring, geothermal gradient, and geothermal flow data to predict the geothermal resources in Akita city and Iwate county in northern Japan [20]. Yousefi selected volcanic rocks, faults, hot springs, hydrothermal alteration zones, magnetic anomalies, and other factors for analysis and calculation [21]. Moghaddam used volcano, fault strike, hot spring, hydrothermal alteration zone, geothermal gradient, and geothermal flow data to build a weight of evidence and credibility function model and used Boolean index-overlay, multilevel index-overlay, and a fuzzy logic model to re-evaluate his findings; he concluded that the fuzzy logic model had the highest accuracy [22,23]. Therefore, it is feasible to quantitatively predict the geothermal anomaly area by evaluating the influence factors closely related to the geothermal area. Among all kinds of geological status and disaster assessment methods, the information model is a commonly used bivariate statistical method which can effectively deal with many factors and difficult-to-quantify natural conditions [24]. However, it is rarely used in the prediction and evaluation of geothermal high-temperature anomaly areas, and the research on the weighted information model is not sufficient.

The purpose of this study is to predict and evaluate the geothermal high-temperature anomaly areas along the Sichuan–Tibet railway. The distribution of geothermal sites is closely related to the Earth's heat flow [25,26], faults, earthquakes [27,28], water, Bouguer gravity, and magnetic anomalies [29]. Therefore, taking the land surface temperature (LST), combined entropy of geological formation, fault density, buffer distance to river, magnetic force, and earthquake peak acceleration as the input impact factors, based on the information model combined with information entropy and weight of evidence theory, an index-overlay, weights of entropy, and evidence information model is established to predict the distribution of geothermal high-temperature anomalies in the study area. This paper will provide suggestions and decision support for the design and construction of the

Sichuan–Tibet railway. The technology and methodology flow chart of this paper is shown in Figure 1.

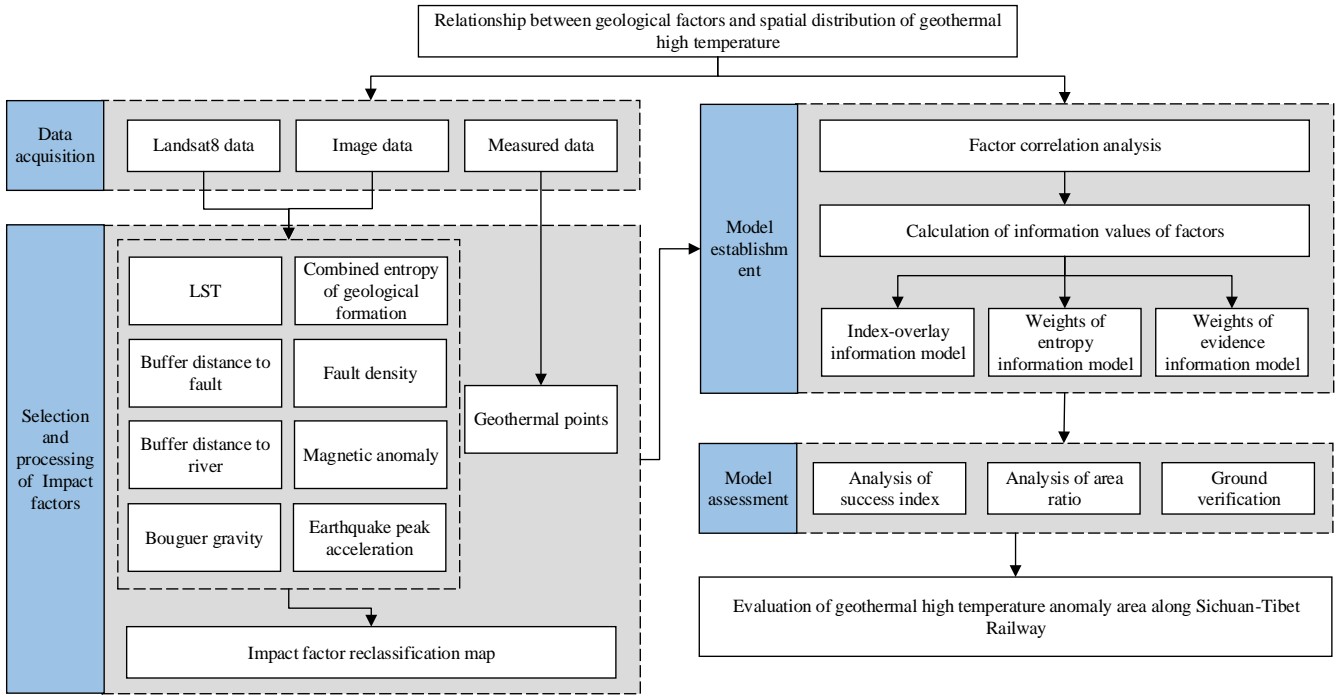

**Figure 1.** The technology and methodology flow chart.

## 2. Study Area

The study area covers the whole Sichuan–Tibet railway, which is located between 93.5° E–103.5° E and 28.5° N–32.5° N in Southwest China. It mainly includes the eastern part of the Tibet Autonomous Region and the western part of Sichuan Province, with a total area of 399,760 km$^2$ (Figure 2a). The area is dominated by the plateau mountain monsoon climate due to the high mountains acting as a barrier to water vapor. The mean annual air temperature is 2.4–12.6 °C and the annual total precipitation is 417–935 mm, with 80% of it occurring between May and September [30].

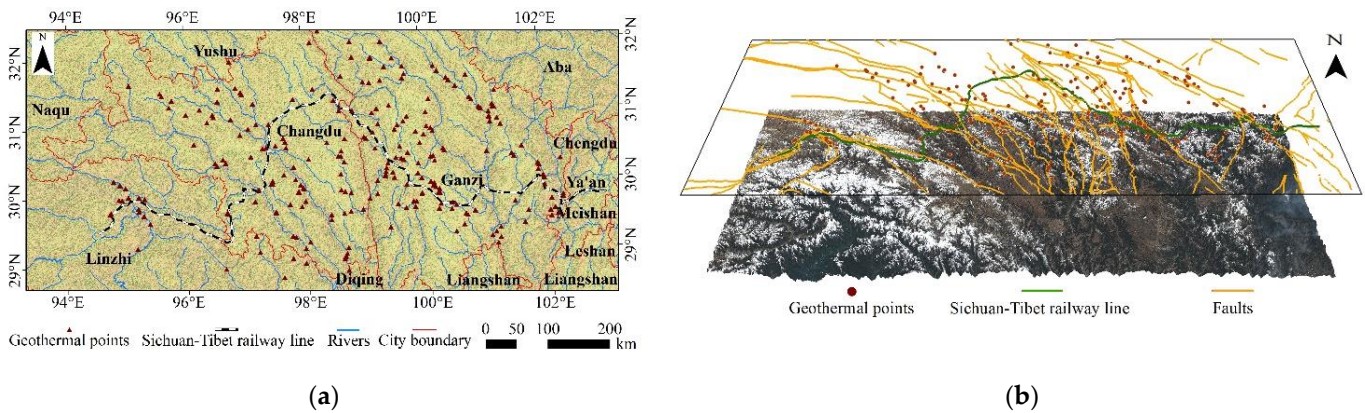

(**a**)                      (**b**)

**Figure 2.** Geographical location and topography of the study area. (**a**) Location of the study area; (**b**) topography of the study area (Landsat8 true color synthetic overlaid DEM; spatial resolution: 30 m; acquisition time: 2018.11–2019.1).

This area has steep terrain, including the Langla, Boshula, Nyainqntanglha, Sejila, Himalayas, and other high mountains, with an average altitude of more than 4000 m. The surface water is intensively distributed, and the Yarlung-Zangbo River, Nujiang River,

Lancang River, and Jinsha River run through the area. High and deep canyons are highly developed, the relative height difference of the canyons is mostly 2000–3000 m, and the maximum is more than 5000 m. The basic topographic conditions are shown in Figure 2b.

The geological conditions in the area are complex, with many types of formation lithology; sedimentary rocks, metamorphic rocks, and intrusive rocks are widely distributed. The Quaternary (Q) strata covered by the surface are mainly sandy soil and gravel soil; the underlying bedrock developed from the Proterozoic to the Cenozoic period. The main active faults include the Lancang River, Nujiang River, Bianba-Luolong, Jiali, and Miling fault zones, which are deep, large, and concentrated. The earthquake activity is frequent and characteristic of a high-intensity earthquake-prone region. Additionally, the study area is located in the Mediterranean-South Asia geothermal anomaly zone, one of the most intense areas of geothermal activity in the Chinese mainland, and the temperature of the outlying hot springs is about 40 °C [31]. To sum up, the geological and geomorphic conditions of the surrounding area of the Sichuan–Tibet railway are intricate and the impact of geothermal high-temperature disasters on the railway design and construction is obvious. Thus, in this area it will be highly useful to carry out geothermal high-temperature anomaly prediction research, so this space range is listed as the study area.

## 3. Methodology

### 3.1. Factor Selection

To examine the genesis and spatial distribution of the geothermal anomalies in this area, our original data included Landsat8 images, a lithology map, a fault map, a river map, a magnetic map, Bouguer gravity data, and earthquake acceleration data. In order to highlight the spatial relationship between these data and geothermal anomalies, they were transformed into a land surface temperature map, a combined entropy of geological formation map, a buffer distance to fault map, a fault density map, a buffer distance to river map, a magnetic map, a Bouguer gravity map, and an earthquake peak acceleration map.

#### 3.1.1. Geothermal Points

The hot and warm springs exposed on the land surface were effective for the characterization of geothermal high-temperature anomalies. These springs and their surrounding areas can be regarded as known geothermal anomaly areas and were thus used in the calculation of subsequent prediction models. The geothermal points in the study area were obtained from field measurements, mainly comprising measurements of exposed hot spring points and drilling points (Figure 3), with a maximum drilling depth of 1575 m. Hot springs measure the temperature of water at a certain depth with portable noncontact thermometers, and drilling points measure the temperature of bedrock at different drilling depths. When the average temperature of a point reaches 30 °C, it is considered as a geothermal anomaly point. There are 249 points in total, with the highest temperature reaching 94 °C and an average temperature of 41 °C (Figure 2).

#### 3.1.2. Land Surface Temperature

The land surface temperature (LST) is mainly influenced by solar radiation and geological activities, which can extract heat information from the Earth's surface, thus providing a basis for the determination of geothermal anomalies [32]. LST information generated from remote sensing images has been successfully applied to the study of geothermal resources and disasters. Therefore, it is feasible to use an LST map as a representative factor influencing terrestrial heat flow. Thermal infrared remote sensing is an important means to obtain LST. Satellite systems that can realize temperature inversion by using thermal infrared bands include MODIS, Landsat8, Landsat7 ETM+, ASTER, NOAA/AVHRR, etc. Among them, Landsat8 data have many advantages: firstly, these data can be obtained directly (http://glovis.usgs.gov/, accessed on 15 August 2020); secondly, thermal infrared band 10 has a high spatial resolution of 30 m and can effectively obtain the fine thermal field landscape and surface temperature anomalies. LST inversion

methods using remote sensing images can be divided into single-channel algorithms [33], multichannel algorithms [34], multiangle algorithms [35], multiphase algorithms [36], and hyperspectral inversion algorithms [37].

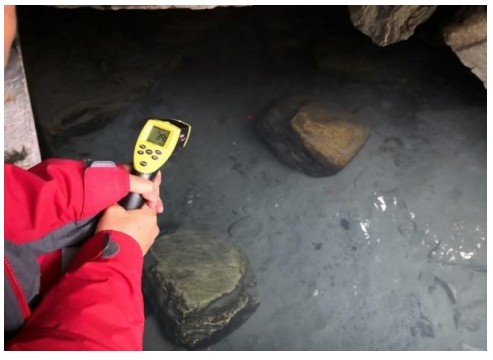
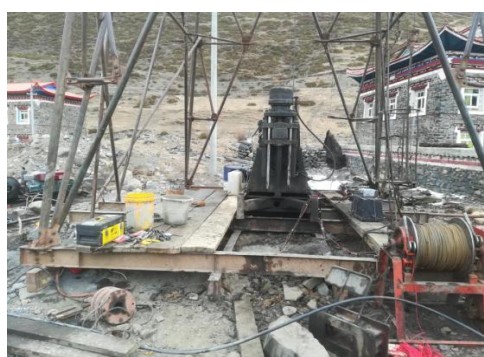

(**a**) Measurement of a hot spring

(**b**) Measurement from drilling

**Figure 3.** Field temperature measurement of geothermal points.

Based on the radiative transfer equation, Qin et al. established a single-window algorithm for LST inversion using only one thermal infrared band [38]. The algorithm only needs three parameters—namely, surface emissivity, atmospheric transmittance, and average atmospheric temperature—to retrieve LST. For the surface emissivity, the normalized difference vegetation index (NDVI) threshold method was used to calculate it. In the scale of the Landsat8 satellite, the Earth's surface can be roughly regarded as being covered by four types of ground features: water, vegetation, buildings, and soil. According to the different values of NDVI, we can determine the proportions of various ground features in the mixed pixels to obtain the surface emissivity. The atmospheric transmittance, according to previous studies, mainly depends on the atmospheric water vapor content, and w can be obtained through the MOD05_L2 product. For mid-latitude winter, the relationship between atmospheric transmittance and atmospheric water vapor content is:

$$T = 0.9228 - 0.0735w, \tag{1}$$

where T is the atmospheric transmittance and w is the atmospheric water vapor content.

For the average atmospheric temperature, Qin et al. proposed a linear relation for the approximation of the average atmospheric temperature from the near surface air temperature. For mid-latitude winter, the average atmospheric temperature could be approximated as:

$$T_a = 19.2704 + 0.9112T\_0, \tag{2}$$

where $T_a$ is the average atmospheric temperature and $T\_0$ is the near-surface air temperature.

The single window method is widely used and has a high inversion accuracy, and most of the satellite sensor data with a thermal infrared band can be used for LST inversion by this algorithm [38]. The formula can be expressed as:

$$T_s = \{a(1 - C_i - D_i) + [b(1 - C_i - D_i) + C_i + D_i] T_b - D_i T_a\}/C_i, \tag{3}$$

$$C_i = \varepsilon_i \tau_i, \tag{4}$$

$$D_i = (1 - \tau_i)[1 + \tau_i(1 - \varepsilon_i)], \tag{5}$$

where $T_s$ is the retrieved LST; $T_b$ is the brightness temperature (k) obtained by the sensor; $T_a$ is the average atmospheric temperature (k); a and b are the linear regression coefficients, which are related to the temperature range of the study area; C and D are intermediate variables; $\varepsilon$ is the surface emissivity; and $\tau$ is the atmospheric transmittance [39].

The inversion results of LST in plateau and mountainous areas are significantly affected by solar radiation heating, which covers the effect of geological activities on LST

changes and needs to be eliminated by topographic correction. Empirical statistics is a commonly used terrain correction method. There is a certain linear relationship between the solar radiation and the radiance value of image pixels. Through regression analysis, the linear relationship between them was established, and then the radiation energy received by the slope pixels was corrected to the horizontal position through the regression relationship so as to eliminate the effect of terrain. The calculation formulae are as follows:

$$\cos(i) = \cos(z)\cos(S) + \sin(z)\sin(S)\cos(\Phi_x - \Phi_n), \tag{6}$$

$$L_T = m\cos(i) + b, \tag{7}$$

$$L_H = L_T - [m\cos(i) + b] + \overline{L_T}, \tag{8}$$

where i is the effective incidence angle of the sun; z is the solar zenith angle; $\Phi_x$ is the sun azimuth angle; S is the pixel slope angle; $\Phi_n$ is the pixel aspect angle; $L_T$ is the radiance value of the ground object before correction; m and B are the parameters obtained by regression analysis; $L_H$ is the radiance value after correction; $\overline{L_T}$ is the radiance value of ground features in a flat area.

The study area needs 15–20 images for full coverage; a total of 104 winter Landsat8 images were obtained in the periods November 2013–February 2014, December 2014–January 2015, December 2015–February 2016, November 2016–January 2017, December 2017–February 2018, and November 2018–January 2019. LST values were obtained by using the single-window algorithm, and terrain correction was carried out. The multiyear average LST (Figure 4a) was calculated as the input data of the geothermal anomaly prediction model. Since the study only focuses on high-temperature anomalies, the lower threshold of LST was set to −10 °C. It can be seen that the low-temperature anomalies were mainly concentrated in the mountains with high altitudes, while the high temperature anomalies were distributed in the valleys and plains.

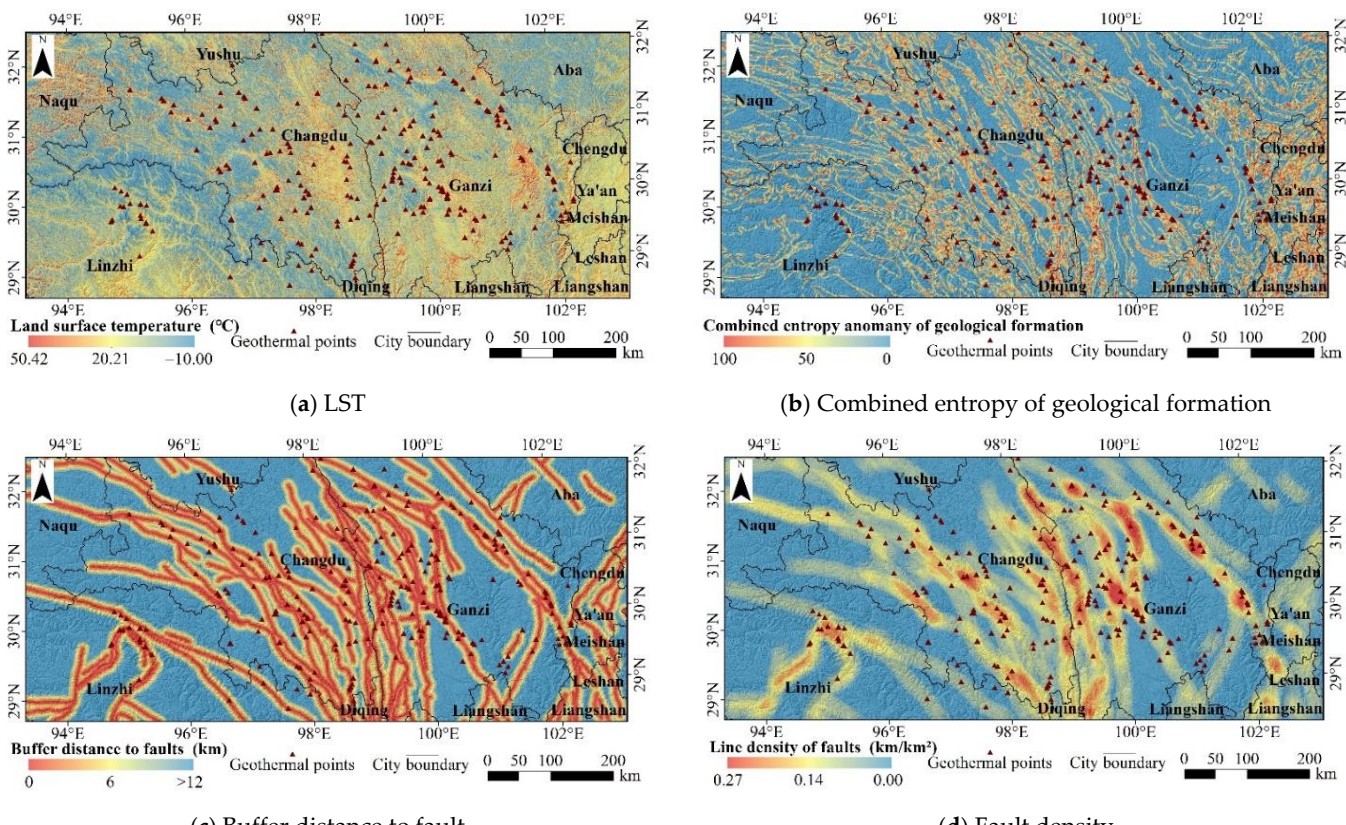

(**a**) LST

(**b**) Combined entropy of geological formation

(**c**) Buffer distance to fault

(**d**) Fault density

**Figure 4.** *Cont.*

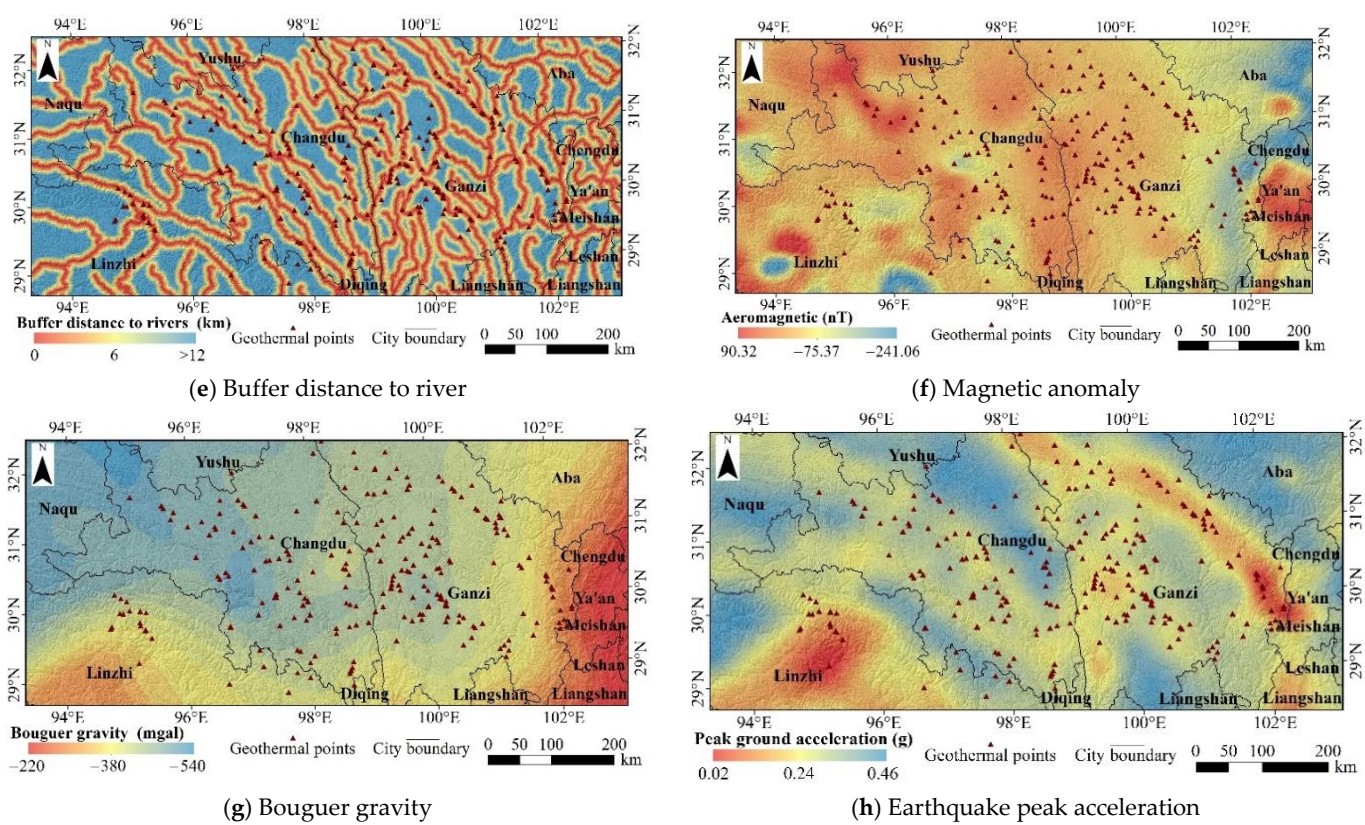

(**e**) Buffer distance to river

(**f**) Magnetic anomaly

(**g**) Bouguer gravity

(**h**) Earthquake peak acceleration

**Figure 4.** Factor maps.

### 3.1.3. Combined Entropy of Geological Formation

The comprehensive variable constructed by relative entropy can reflect the lithologic combination characteristics in the unit area, which can quantitatively characterize the degree of homogeneity or variation in a geological structure in a certain range [40,41]. Entropy anomaly can be used as an important index to reveal geological anomaly. The combined entropy of geological formation is the entropy anomaly of various geological bodies or different attributes of the same geological body in terms of unit area or volume, which can be calculated using the lithology data (http://geocloud.cgs.gov.cn/, accessed on 2 October 2020). In a certain range, the higher the entropy value is, the higher the variation degree of a geological structure is, and the higher the possibility of geothermal anomaly is. The lithology data in this study can be divided into 323 categories. The calculation of combined entropy can be divided into the following steps (Figure 5).

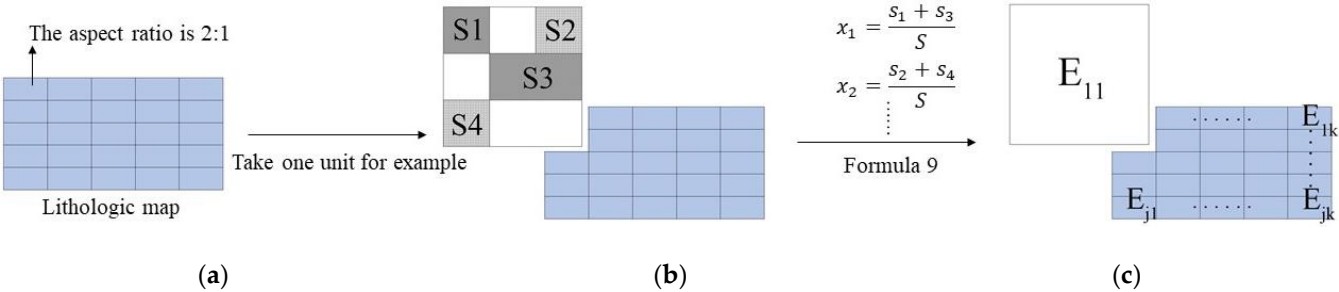

(**a**)

(**b**)

(**c**)

**Figure 5.** The calculation of the combined entropy of geological units. (**a**) Surface pattern in two dimensions; (**b**) surface lithological identification—lithological units are identified by color and pattern, $S_1$ and $S_3$ are surfaces belonging to the same lithological unit, $S_2$ and $S_4$ correspond to another lithological unit; (**c**) the calculation of combined entropy.

First, the lithologic map needs to be divided into several grid units. The long axis direction, shape, and size are the elements of unit division. The direction of the long axis of the grid unit should be consistent with the direction of the regional tectonic line. The main fault direction of the study area is southeast, so the long axis of the unit is also southeast. The shape of the grid unit should correspond to the shape of the stratum. The stratum in the area has an obvious strike, and the ratio of width to length is about 1/2–1/3, so the width/length ratio of the grid unit is 1/2. After determining the grid units, each independent lithology area in the unit is calculated, then the sum of their area in the unit is calculated and the ratio $x_i$ (i = 1,2,3 . . . , n) of each lithology area in the unit to the unit area is calculated. On this basis, the combined entropy of geological formation can be evaluated by the following formula:

$$E_{jk} = - \sum_{i=1}^{n} x_i \ln x_i / \ln n, \tag{9}$$

where n is the lithology type existing in the grid unit and j and K are the row number and column number of the unit. The calculated combined entropy of geological formation is shown in Figure 4b. The high values for the combination entropy are concentrated in the linear area with sharp changes in lithology, which has a high spatial correlation with the pattern of geological faults.

### 3.1.4. Buffer Distance to Fault and Fault Density

It is well known that deeper faults form seepage channels where groundwater can flow towards the deep crust and reach high temperatures. When the heated water returns to the surface again through shallow faults, geothermal occurrences are possible [42]. Based on the fault data (Figure 2b, http://geocloud.cgs.gov.cn/, accessed on 5 October 2020), the buffer distance to fault and fault density can be used to determine the relationship between geothermal points and active thermal geology. In the buffer distance map, each pixel contains the vertical distance information from the nearest fault. Figure 4c clearly shows that there are 208 geothermal points in total, accounting for 83.53% of the total number of points, which are distributed within 4 km from the fault. It can be seen from Figure 4d that there are 154 geothermal points, accounting for 61.85% of all points, which are distributed in the area with a fault density greater than 0.05 km/km$^2$.

### 3.1.5. Buffer Distance to River

The different migration modes of water channels have different effects on geothermal formation. The downward migration of groundwater leads to a decrease in rock temperature deep underground, where it is not easy to generate geothermal high-temperature anomalies; when the groundwater heated by the surrounding rock circulates upward under certain geological conditions, it will cause the local temperature of surrounding rock to rise and form a high-temperature anomaly in the shallow part of the surface [43]. Therefore, it is feasible to use the distance to river system as the influencing factor for geothermal areas. The data of rivers in the study area with a scale of 1:3 million were divided into 3 classes, including 3-level rivers (basin area of 1–10 thousand km$^2$), 4-level rivers (basin area of 0.1–1 thousand km$^2$), and 5-level rivers (basin area <0.1 thousand km$^2$) [44]. In order to study the spatial relationship between the river system and geothermal activity, the river was converted into buffer distance (Figure 4e). Statistics show that 85.54% of geothermal points (213) are distributed within a range of less than 4 km from the river, indicating a close relationship between river and geothermal anomalies. The abundant water resources in the study area provide a material and energy basis for the occurrence of geothermal high-temperature anomalies.

### 3.1.6. Magnetic Anomaly

Magnetic anomalies are often used to indicate the active area of underground hot water and the areas with significant changes in tectonic stress, so the magnetic layer can be used as the influence factor [45]. In theory, lower aeromagnetic values are closely related to geothermal area; in the range of geothermal water activities, the magnetic properties of

rocks are reduced due to thermal alteration; on the other hand, under the action of tectonic stress, the magnetic properties of rocks along the stress direction will be weakened, so the magnetic properties of rocks in the tectonic fracture zone will also be reduced. According to the magnetic anomaly map (Figure 4f) [46], most of the geothermal points are distributed in the medium magnetic region ($-70$--$-80$ nT), and the points with high magnetism (>100 nT) and low magnetism (<$-50$ nT) are relatively few.

### 3.1.7. Bouguer Gravity

Bouguer gravity anomaly is the area where the density of crustal material changes sharply along the horizontal direction, which is a sign of the existence of a graben system [47]. The reason for the formation of this is the uneven distribution of underground rock mass and mineral density, or a density difference between the geological body and surrounding rock. Bouguer gravity anomaly can be used to understand regional structures and delineate large fault structures and possible hydrothermal activities. The Bouguer gravity anomaly can reflect the variation of crustal thickness in different areas, and the rock density decreases with the increase in temperature. The distribution gradient of Bouguer gravity anomaly is consistent with the distribution of hot springs, so it is regarded as one of the impact factors. A Bouguer gravity map is shown in Figure 4g.

### 3.1.8. Earthquake Peak Acceleration

In the seismicity area and active fault zone, due to the high permeability and circulation of water the hydrothermal activity is closely related to the seismic area. The seismicity can be studied by means of the intensity of the earthquake, and then the spatial relationship between the hydrothermal activities in the region can be found [48,49]. The index of earthquake peak acceleration is used to represent the intensity of seismic activity. From the map of earthquake peak acceleration (Figure 4h, http://geocloud.cgs.gov.cn/, accessed on 12 October 2020), we can see that there are more geothermal points distributed in the areas with medium and high earthquake peak acceleration.

### *3.2. Factor Reclassification and Independence*
### 3.2.1. Factor Reclassification

The reclassification of impact factors is an important part of the subsequent model prediction, and the appropriate threshold is needed to obtain the reasonable impact factor classification map. Generally speaking, the more classes the factor map divides, the more objective and reliable the analysis results are. However, too many classes will increase the computational burden. Therefore, in the process of reclassification, using the natural breakpoint method, each factor map determines seven classes, and the area between the intervals is neither too large nor too small. In this way, all the factors obtain the threshold for the reclassification of impact factors (Table 1). Additionally, when using a square grid as the evaluation unit, the size of the grid should be considered. In theory, the smaller the grid, the more similar the geological environment of each point in the grid, and the higher the accuracy of evaluation and prediction. However, if the size is too small, the calculation efficiency will be seriously reduced. Therefore, when selecting the grid size, the identity of geological environment in the unit, the suitability of the size proportion of the study area, and the computer processing ability are taken into account. According to the experiment, Tang et al. put forward the corresponding grid size selection standard (Table 2) [50]. Combined with the scale of the study area, this study selects 100 × 100 m as the grid size and resamples the factor map after reclassification (Figure 6).

**Table 1.** Reclassification threshold for impact factor maps.

| Impact Factors | Class | | | | | | |
|---|---|---|---|---|---|---|---|
| | 1 | 2 | 3 | 4 | 5 | 6 | 7 |
| LST (°C) | −10.00–6.21 | −6.21–0.69 | 0.69–6.83 | 6.83–11.75 | 11.75–16.02 | 16.02–23.54 | 23.54–50.42 |
| Combined entropy of geological formation | 0–10 | 10–20 | 20–40 | 40–50 | 50–60 | 60–80 | 80–100 |
| Buffer distance to fault (km) | 0–2 | 2–4 | 4–6 | 6–8 | 8–10 | 10–12 | >12 |
| Fault density (km/km²) | 0.000–0.019 | 0.019–0.047 | 0.047–0.069 | 0.069–0.092 | 0.092–0.117 | 0.117–0.148 | 0.148–0.266 |
| Buffer distance to river (km) | 0–2 | 2–4 | 4–6 | 6–8 | 8–10 | 10–12 | >12 |
| Magnetic anomaly (nT) | −241–−163 | −163–−115 | −115–−75 | −75–−50 | −50–−26 | −26–8 | 8–100 |
| Bouguer gravity (mgal) | −540–−520 | −520–−480 | −480–−440 | −440–−400 | −400–−340 | −340–−300 | −300–−220 |
| Earthquake peak acceleration (g). | 0.02–0.08 | 0.08–0.14 | 0.14–0.20 | 0.20–0.24 | 0.24–0.30 | 0.30–0.34 | 0.34–0.40 |

**Table 2.** Grid size selection standard.

| Study Area (km²) | Scale | Grid Size (m) |
|---|---|---|
| ≥100,000 | ≤1:250,000 | ≥100 × 100 |
| 10,000–100,000 | 1:250,000–1:100,000 | ≥50 × 50 |
| 1000–10,000 | 1:100,000–1:50,000 | ≥25 × 25 |
| <10,000 | >1:50,000 | ≥5 × 5 |

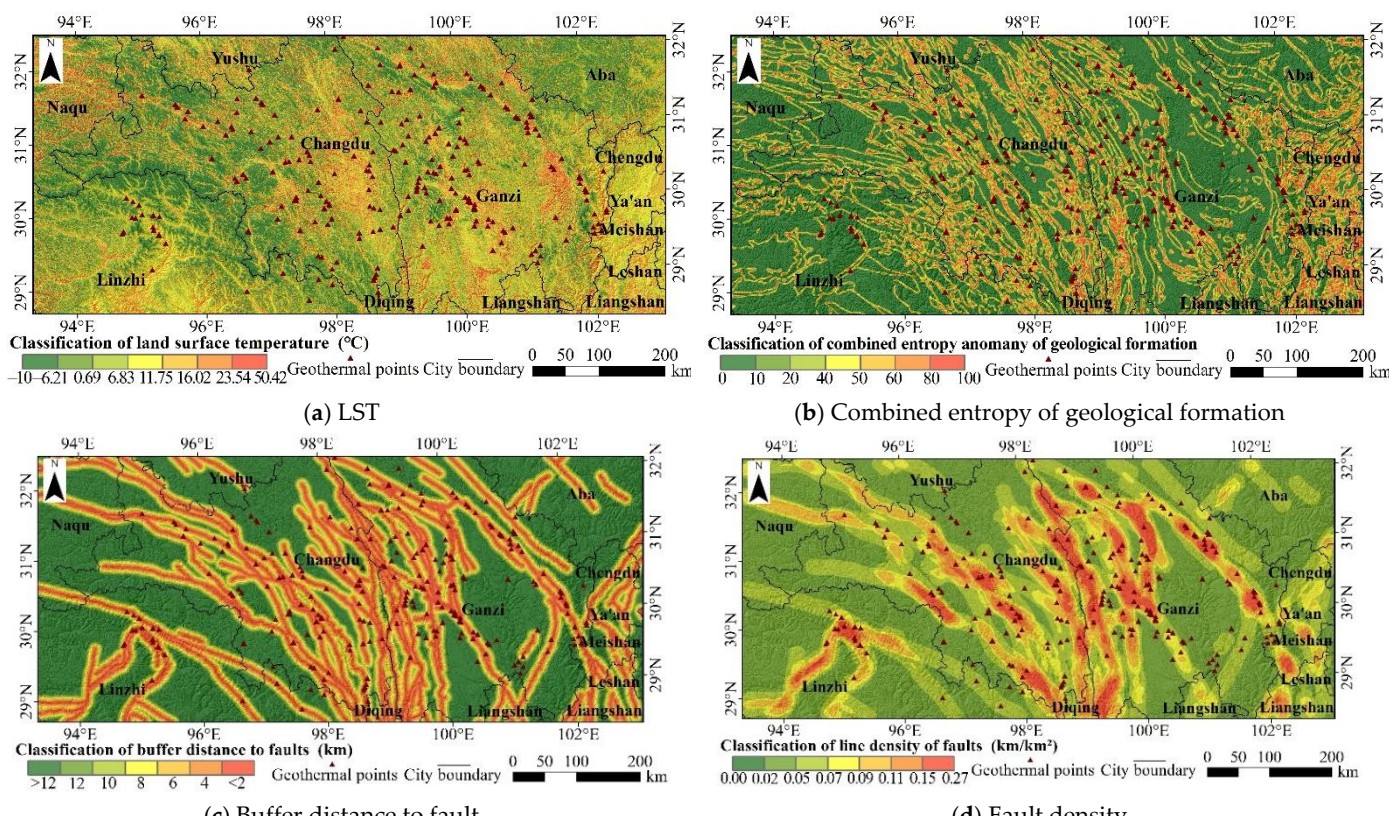

(**a**) LST

(**b**) Combined entropy of geological formation

(**c**) Buffer distance to fault

(**d**) Fault density

**Figure 6.** *Cont.*

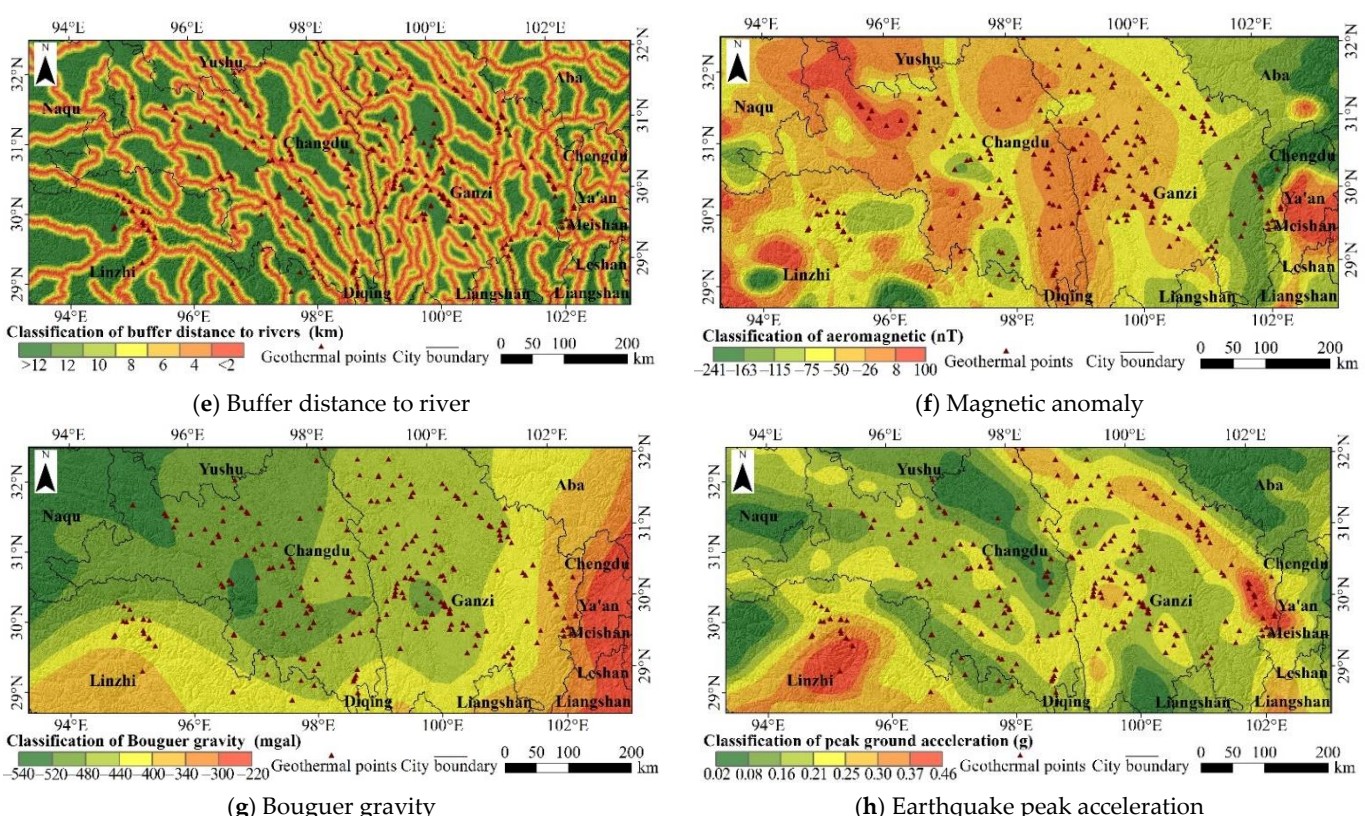

**Figure 6.** Factor reclassification maps.

### 3.2.2. Independence Test

The establishment of an information model needs a strict assumption—that is, that the selected influence factors are independent of each other [51]. In this paper, based on the results of the factor reclassification and factor analysis, the independence was judged—that is, the correlation coefficient was used to measure the correlation of factors, the correlation between each factor was analyzed by calculating the covariance and correlation matrix, and the factors with a large correlation were excluded to ensure the effectiveness of the model application. When the correlation coefficient of each factor satisfies $|R| \leq 0.3$, it can be considered as a weak correlation or no correlation [52]. From Table 3, it can be seen that the correlation between he Bouguer gravity and magnetic map is high, so the Bouguer gravity factor is excluded; the buffer distance to the fault and fault density are generated by fault data, so the correlation is the highest. When $|R|$ reaches 0.771, the buffer distance to the fault factor is excluded. Finally, the LST, combined entropy of geological formation, fault density, buffer distance to rivers, and magnetic and earthquake peak acceleration are retained as the influencing factors in the model.

### 3.3. Model Establishment

Based on the information model combined with a variety of objective weighting methods, the index-overlay information model, weights of entropy information model, and weights of evidence information model are proposed. These models are used to predict and evaluate the geothermal anomaly in the study area.

#### 3.3.1. Information Model

The main idea of the information model applied to the evaluation of geothermal anomaly area is to combine the geological conditions closely related to the formation of high-temperature geothermal anomalies, convert the values of these influencing factors into information values, and calculate the information value provided by each influence factor

on the generation of high-temperature geothermal anomalies to analyze the relationship between the two so as to combine the geological geomorphology, remote sensing information, and other elements to build a prediction model for geothermal high-temperature anomalies [24]. The information model is a bivariate statistical analysis method, geothermal anomaly is affected by many factors that will form a kind of "optimal combination of factors" to cause the occurrence of a geothermal anomaly [53]. Its formation is related to the quantity and quality of influencing factors; the greater the amount of information there is, the greater the possibility of forming a geothermal anomaly.

**Table 3.** Correlation of impact factor maps.

| Impact Factors | LST | Combined Entropy | Buffer Distance to Fault | Fault Density | Buffer Distance to River | Magnetic Anomaly | Bouguer Gravity | Earthquake Peak Acceleration |
|---|---|---|---|---|---|---|---|---|
| LST | 1.000 | −0.062 | 0.127 | −0.152 | 0.038 | 0.053 | 0.089 | −0.006 |
| Combined entropy | −0.062 | 1.000 | −0.158 | 0.175 | −0.157 | 0.035 | −0.079 | −0.037 |
| Buffer distance to fault | 0.127 | −0.158 | 1.000 | −0.771 | 0.064 | 0.171 | −0.021 | 0.263 |
| Fault density | −0.152 | 0.175 | −0.771 | 1.000 | −0.085 | −0.160 | −0.031 | −0.246 |
| Buffer distance to river | 0.038 | −0.157 | 0.064 | −0.085 | 1.000 | −0.034 | 0.150 | 0.051 |
| Magnetic anomaly | 0.053 | 0.035 | 0.171 | −0.160 | −0.034 | 1.000 | −0.406 | −0.016 |
| Bouguer gravity | 0.089 | −0.079 | −0.021 | −0.031 | 0.150 | −0.406 | 1.000 | 0.292 |
| Earthquake peak acceleration | −0.006 | −0.037 | 0.263 | −0.246 | 0.051 | −0.016 | 0.292 | 1.000 |

When the information model is used to evaluate the geothermal anomaly area, n factors affecting geothermal energy are selected and expressed as $X_i$ (i = 1,2,3 ... , n). The geothermal information provided by single factor $X_i$ is $I_i$ and the total geothermal information provided by n factors is I. $I_i$ and I are expressed as follows:

$$I_i = \log_2 \frac{P[Y/X_i]}{P(Y)}, \tag{10}$$

$$I = \log_2 \frac{P[Y/X_1 X_2 \cdots X_n]}{P(Y)}. \tag{11}$$

Based on $3.99 \times 10^7$ grids with equal area of $100 \times 100$ m in the study area, the amount of information provided by each class of influencing factor $X_i$ (i = 1,2,3 ... , n) to geothermal anomaly is $I_i$, which can be expressed as follows:

$$I_i = \log_2 \frac{N_{ij}/N}{S_{ij}/S}, \tag{12}$$

where S is the total number of grids in the study area, $S_{ij}$ is the number of grid units with $X_{ij}$ of j class, n is the total number of geothermal points in the study area, and $N_{ij}$ is the number of geothermal points of $X_{ij}$ with class J.

The total information value I of the information model is as follows:

$$I = \sum_{i=1}^{n} I_i = \sum_{i=1}^{n} \log_2 \frac{N_{ij}/N}{S_{ij}/S}, \tag{13}$$

where n is the number of influence factors.

After calculation, the information values of each factor are shown in Table 4. The + and −information values indicate the positive and negative correlation with geothermal high-temperature anomalies. For example, the information value of the LST 1st class is −3.145, which indicates that this class will greatly lessen the occurrence of geothermal high-temperature anomalies, while the information of the seventh class is 0.926, and the

geothermal temperature of this class will promote the occurrence of geothermal anomaly. With the increase in the series of LST, the combined entropy of geological formation, the fault density, and the earthquake peak acceleration, the amount of information provided increases, which indicates that these factors have a positive correlation with geothermal high-temperature anomalies as a whole. The larger their values are, the more likely it is that a geothermal high-temperature anomaly will occur. Among them, the positive value of the information provided by LST, fault density, and earthquake peak acceleration is very prominent, up to 1.203. There is a negative correlation between the buffer distance to rivers and magnetic anomalies and geothermal high-temperature anomalies. With the increase in their values, the possibility of geothermal anomaly decreases.

**Table 4.** Information values for impact factors.

| Impact factors | Class | Points | Grids | $I_{ij}$ | $H_i$ | $W_i$ |
|---|---|---|---|---|---|---|
| LST | 1 | 2 | 7,459,893 | −3.145 | | |
| | 2 | 2 | 6,526,266 | −3.012 | | |
| | 3 | 26 | 6,864,561 | −0.497 | | |
| | 4 | 55 | 7,272,770 | 0.194 | 0.803 | 2.462 |
| | 5 | 85 | 6,157,522 | 0.796 | | |
| | 6 | 51 | 3,924,683 | 0.736 | | |
| | 7 | 28 | 1,781,793 | 0.926 | | |
| Combined entropy of geological formation | 1 | 53 | 21,710,503 | −0.936 | | |
| | 2 | 18 | 2,378,633 | 0.195 | | |
| | 3 | 42 | 4,703,793 | 0.360 | | |
| | 4 | 26 | 2,538,350 | 0.498 | 0.955 | 0.568 |
| | 5 | 37 | 2,590,893 | 0.830 | | |
| | 6 | 50 | 4,408,954 | 0.599 | | |
| | 7 | 23 | 1,656,362 | 0.802 | | |
| Fault density | 1 | 40 | 5,387,518 | 0.176 | | |
| | 2 | 55 | 4,568,425 | 0.659 | | |
| | 3 | 34 | 3,905,275 | 0.335 | | |
| | 4 | 44 | 3,325,438 | 0.754 | 0.932 | 0.847 |
| | 5 | 33 | 2,855,477 | 0.618 | | |
| | 6 | 30 | 2,452,899 | 0.675 | | |
| | 7 | 13 | 17,492,456 | −2.126 | | |
| Buffer distance to river | 1 | 112 | 5,920,266 | 1.111 | | |
| | 2 | 36 | 5,608,909 | 0.030 | | |
| | 3 | 22 | 5,273,756 | −0.401 | | |
| | 4 | 20 | 4,772,597 | −0.396 | 0.869 | 1.647 |
| | 5 | 18 | 4,150,553 | −0.362 | | |
| | 6 | 13 | 3,479,987 | −0.511 | | |
| | 7 | 28 | 10,781,420 | −0.875 | | |
| Magnetic anomaly | 1 | 5 | 1,393,066 | −0.551 | | |
| | 2 | 9 | 2,079,263 | −0.364 | | |
| | 3 | 20 | 5,555,534 | −0.548 | | |
| | 4 | 70 | 9,336,800 | 0.186 | 0.964 | 0.457 |
| | 5 | 88 | 11,748,298 | 0.185 | | |
| | 6 | 52 | 7,902,464 | 0.055 | | |
| | 7 | 5 | 1,972,063 | −0.899 | | |
| Earthquake peak acceleration | 1 | 7 | 4,149,900 | −1.306 | | |
| | 2 | 13 | 6,921,418 | −1.199 | | |
| | 3 | 69 | 13,669,294 | −0.210 | | |
| | 4 | 83 | 9,371,586 | 0.352 | 0.980 | 0.254 |
| | 5 | 52 | 3,792,104 | 0.789 | | |
| | 6 | 23 | 1,108,662 | 1.203 | | |
| | 7 | 2 | 974,524 | −1.110 | | |

### 3.3.2. Index-Overlay Information Model

Due to the difference in the quality and quantity of geological and geophysical data, the contribution of influencing factors to the formation of geothermal anomaly is different. However, the traditional information model simply superimposes the information values of each factor, and it is difficult to reflect the distinction of the factors' influence on geothermal anomalies. Therefore, several objective weighting methods are used to establish the weighted information model and discuss the different contribution degrees of each factor. The index-overlay is a method that directly overlays the information value of the factors [54], and its expression is as follows:

$$\text{Index} = \sum_{i=1}^{n} W_i \times I_i = \sum_{i=1}^{n} W_i \times \log_2 \frac{N_{ij}/N}{S_{ij}/S}, \tag{14}$$

where Index is the superposition value of weighted information and $W_i$ is the weight of each influence factor. In the index-overlay model, all the factors have equal effects, $W_i = 0.167$.

### 3.3.3. Weights of Entropy Information Model

Using the theory of information entropy, objective weight can be obtained to reflect the importance of factors. According to the theory of information entropy, the smaller the entropy value of a factor is, the greater the change in the density of geothermal points in each classification is, and the higher the corresponding weight is, the greater the effect of this factor will be on the prediction and evaluation objectives [55]. The calculation formula for geothermal activity is:

$$d_{ij} = \frac{N_{ij}}{S_{ij}}, \tag{15}$$

where $d_{ij}$ is the geothermal occurrence rate of the j level in the ith factor layer and the meanings of $N_{ij}$ and $S_{ij}$ are consistent with Formula 10.

The formula used to calculate the normalization value for points can be expressed as follows:

$$K_{ij} = \frac{d_{ij}}{\sum_{j=1}^{n} d_{ij}}, \tag{16}$$

where $K_{ij}$ is the normalization value for the jth class of the ith map.

According to the informational entropy theory, the entropy is:

$$H_i = -\frac{\sum_{j=1}^{m} K_{ij} \ln K_{ij}}{\ln m}, \tag{17}$$

where $H_i$ is the theoretical value for the ith map. If $K_{ij} = 0$, then $\ln K_{ij} = 0$ and $H_i = 0$.

The weight of the ith map is:

$$W_{ij} = \frac{(1 - H_i) \times n}{n - \sum_{i=1}^{n} H_i}. \tag{18}$$

The weights of entropy information mode can be expressed as follows:

$$\text{Entropy} = \sum_{i=1}^{n} W_i \times I_i = \sum_{i=1}^{n} W_i \times \log_2 \frac{N_{ij}/N}{S_{ij}/S}, \tag{19}$$

where Entropy is the calculated result for the pixels and $W_i$ is the weight of the ith map (Table 4).

### 3.3.4. Weights of Evidence Information Model

The weight of evidence is a quantitative evaluation method based on Bayesian statistics. Through the spatial correlation analysis of some geothermal anomalies that have

occurred and the related factors affecting the formation of geothermal anomalies, the contribution rate (weight) of each factor to the geothermal anomaly can be determined and used to calculate the possibility of geothermal anomaly. Finally, each factor is given the corresponding weight, and the geothermal anomaly is obtained by superposition. Its advantage is that the interpretation of comprehensive weights is relatively intuitive and easy to understand. By introducing the weight of evidence into the information model, more objective and accurate factor weights can be obtained, which makes the prediction results more accurate [56].

The probability of the random occurrence of training points in the whole study area is defined as the prior probability and is applicable to the whole research area, and its calculation formula is as follows:

$$P(d) = \frac{M}{T},$$ (20)

where P(d) is an a priori probability, M is the number of evaluation units with geothermal points, and T is the total number of evaluation units in the study area.

The weight of the evidence layer can be defined as:

$$W_{ij}^k = \begin{cases} \ln\dfrac{P(e_{ij}|d)}{P(e_{ij}|\overline{d})}, & k = + \\ \ln\dfrac{P(\overline{e_{ij}}|d)}{P(\overline{e_{ij}}|\overline{d})}, & k = - \\ 0, & k = 0 \end{cases}$$ (21)

where P(x|y) is the conditional probability of X phenomenon when y phenomenon occurs; $e_{ij}$ is the number of grids in J class of the ith evidence factor; d is the number of grids with geothermal anomaly; $\overline{e_{ij}}$ is that j class of the ith evidence factor does not occur; $\overline{d}$ is no geothermal anomaly; k is the state of the ith evidence layer in the unit, k = + represents positive correlation weight, and k = − is negative correlation weight. When the positive weight is greater than 0 or the negative weight is less than 0, it means that the factor is positively correlated with the geothermal anomaly. When the positive weight is less than 0 or the negative weight is greater than 0, this means that the factor has little influence on the geothermal anomaly. When the positive weight or negative weight is 0, this means that the factor is not related to geothermal anomaly.

Contrast (C) represents the correlation between the evidence and training points. The greater the C is, the closer the correlation is. Positive is defined as favorable to the target, and negative is unfavorable to the target. The contrast can be used as the final weight and combined with the information value. The calculation formula of C is as follows:

$$C_{ij} = W_{ij}^+ - W_{ij}^-.$$ (22)

The weights of evidence information mode can be expressed as follows:

$$\text{Evidence} = \sum_{i=1}^{n} C_i \times I_i = \sum_{i=1}^{n} C_i \times \log_2 \frac{N_{ij}/N}{S_{ij}/S}$$ (23)

where evidence is the predicted value of the weight of evidence method and $C_i$ is the weight of the ith factor (Table 5).

**Table 5.** Values of the weights of evidence for impact factors.

| Impact Factors | Class | $W_{ij}^+$ | $W_{ij}^-$ | $C_{ij}$ | $C_i$ |
|---|---|---|---|---|---|
| LST | 1 | −0.346 | 0.198 | −0.544 | |
| | 2 | −0.312 | 0.170 | −0.482 | |
| | 3 | −0.498 | 0.078 | −0.576 | |
| | 4 | 0.194 | −0.049 | 0.243 | 3.547 |
| | 5 | 0.896 | −0.350 | 1.246 | |
| | 6 | 1.535 | −0.226 | 1.761 | |
| | 7 | 1.725 | −0.174 | 1.899 | |
| Combined entropy of geological formation | 1 | −0.924 | 0.529 | −1.453 | |
| | 2 | 0.195 | −0.014 | 0.208 | |
| | 3 | 0.360 | −0.060 | 0.420 | |
| | 4 | 0.497 | −0.045 | 0.542 | 2.203 |
| | 5 | 0.830 | −0.094 | 0.924 | |
| | 6 | 0.599 | −0.107 | 0.706 | |
| | 7 | 0.802 | −0.055 | 0.856 | |
| Fault density | 1 | −1.021 | 0.416 | −1.437 | |
| | 2 | −0.160 | 0.050 | −0.211 | |
| | 3 | 0.283 | −0.038 | 0.321 | |
| | 4 | 0.586 | −0.091 | 0.677 | 3.383 |
| | 5 | 0.978 | −0.091 | 1.069 | |
| | 6 | 1.310 | −0.095 | 1.406 | |
| | 7 | 1.515 | −0.042 | 1.557 | |
| Buffer distance to river | 1 | 1.111 | −0.737 | 1.848 | |
| | 2 | 0.730 | −0.650 | 1.380 | |
| | 3 | −0.101 | 0.049 | −0.150 | |
| | 4 | −0.096 | 0.043 | −0.140 | 1.324 |
| | 5 | −0.262 | 0.035 | −0.297 | |
| | 6 | −0.411 | 0.137 | −0.549 | |
| | 7 | −0.574 | 0.195 | −0.768 | |
| Magnetic anomaly | 1 | −0.251 | 0.015 | −0.266 | |
| | 2 | −0.164 | 0.017 | −0.181 | |
| | 3 | −0.448 | 0.036 | −0.484 | |
| | 4 | 0.685 | −0.464 | 1.149 | 0.862 |
| | 5 | 0.707 | −0.397 | 1.104 | |
| | 6 | 0.455 | −0.314 | 0.769 | |
| | 7 | −0.699 | 0.530 | −1.229 | |
| Earthquake peak acceleration | 1 | −1.443 | 0.089 | −1.532 | |
| | 2 | −0.556 | 0.128 | −0.684 | |
| | 3 | 0.063 | −0.034 | 0.097 | |
| | 4 | 0.290 | −0.109 | 0.398 | 1.183 |
| | 5 | 0.576 | −0.185 | 0.761 | |
| | 6 | 0.623 | −0.278 | 0.901 | |
| | 7 | 0.905 | −0.337 | 1.242 | |

### 3.3.5. Classification Maps of Prediction

In order to compare and analyze the models more intuitively, the information series models need to be further classified. If the model value is lower than 0, this indicates that there is nothing to do with the geothermal high-temperature phenomenon. Therefore, the 1st threshold value is set to 0 and values greater than 0 are divided into three classes according to the natural breakpoint method. The specific segmentation threshold is shown in Table 6, and the classification result is shown in Figure 7. Four in the maps shows the high geothermal anomaly area, where high-temperature anomalies occur frequently, and this is the most serious area affected by geothermal high-temperature disaster; three and two, respectively, indicate the middle and low geothermal anomaly areas, and the possibility and severity of geothermal anomalies in the region are reduced in turn; one indicates that

there is no abnormal area, the probability of occurrence of geothermal anomaly is very low, and it is not affected by geothermal high temperature in this area.

**Table 6.** Threshold of model classification values.

| Model | Class | | | |
|---|---|---|---|---|
| | 1 | 2 | 3 | 4 |
| Index-overlay information model | −1.99–0.00 | 0.00–0.24 | 0.24–0.57 | 0.57–1.32 |
| Weights of entropy information model | −16.21–0.00 | 0.00–6.50 | 6.50–7.21 | 7.21–8.81 |
| Weights of evidence information model | −31.72–0.00 | 0.00–3.84 | 3.84–8.41 | 8.41–18.94 |

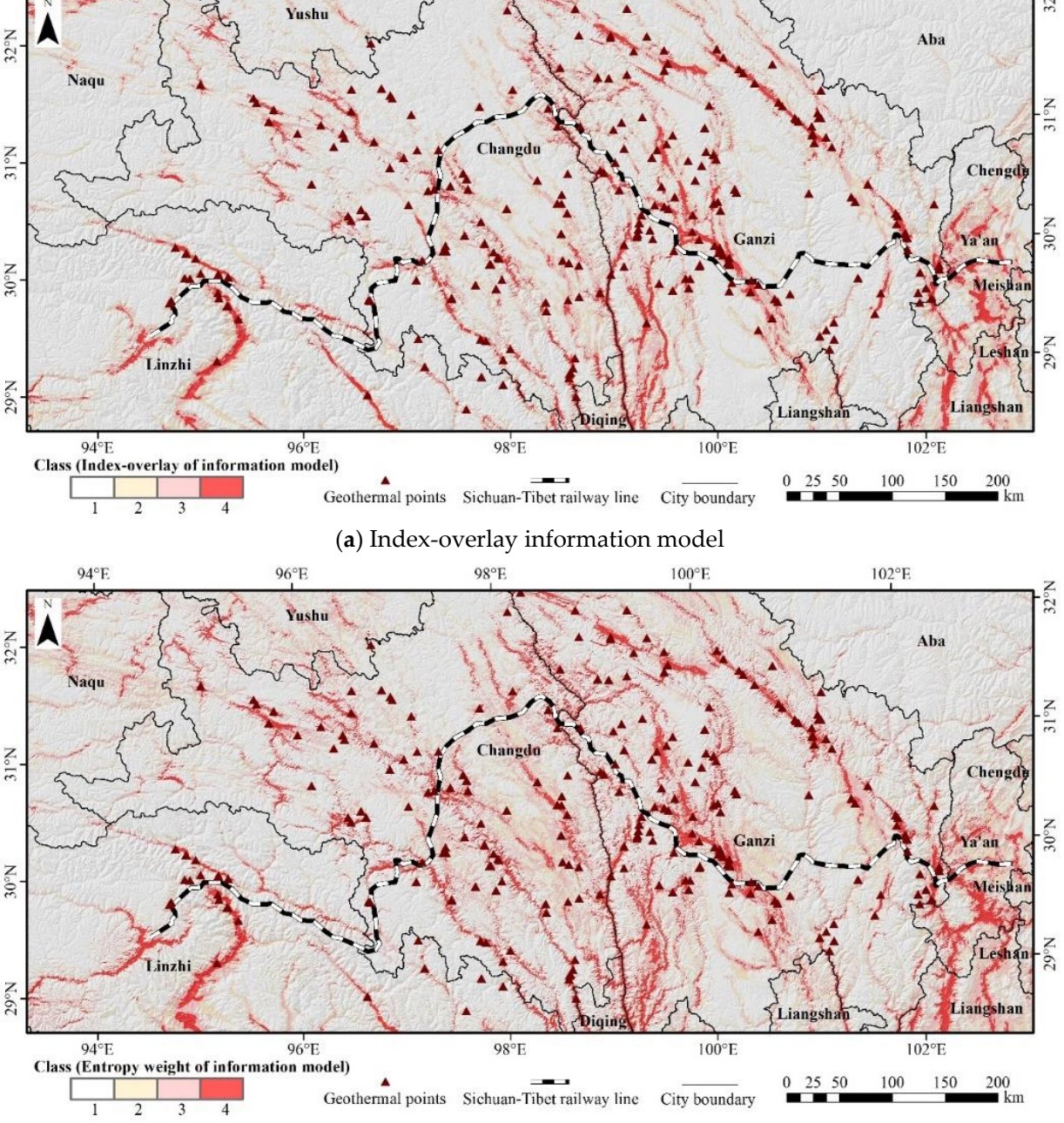

(**a**) Index-overlay information model

(**b**) Weights of entropy information model

**Figure 7.** *Cont.*

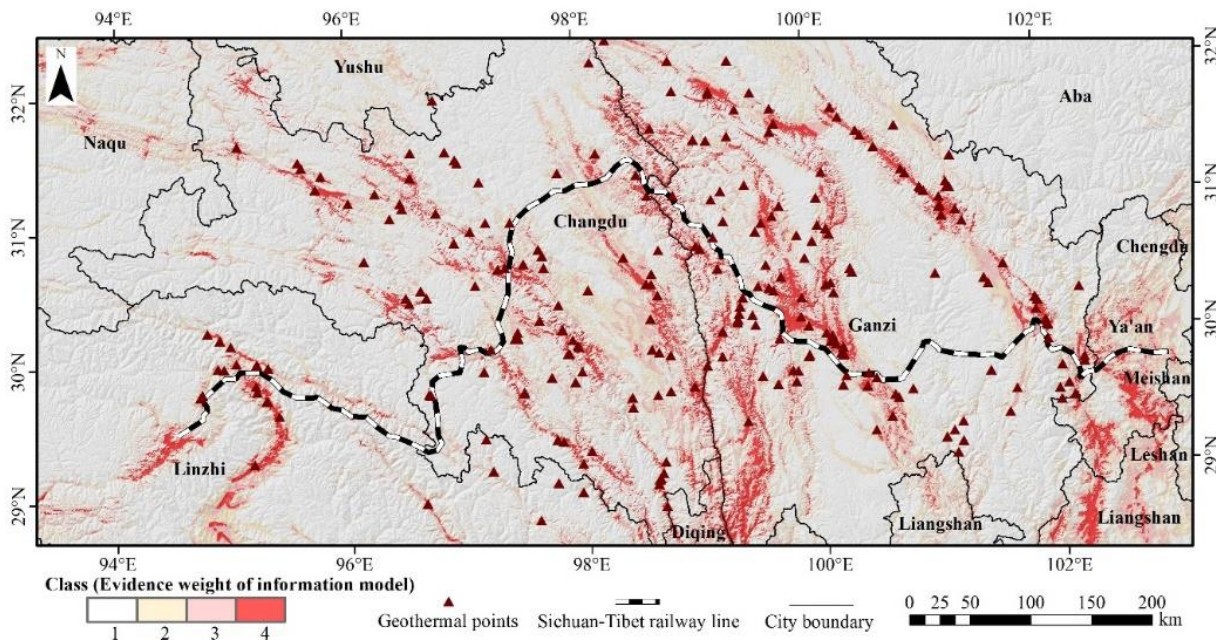

(**c**) Weights of evidence information model

**Figure 7.** Classification maps of prediction.

According to the classification map (Figure 7), the overall prediction results are relatively close, and the geothermal anomaly area is generally in the northwest–southeast direction, which is highly similar to the distribution of faults and water systems. The highly abnormal areas are mainly distributed in the middle of Ganzi and Changdu and the north of Linzhi. The fault structures and rivers are highly developed in these areas, with a high LST and frequent earthquakes. Among them, the geothermal anomaly areas in Ganzi and Linzhi coincide with the Sichuan–Tibet railway line greatly, which has a severe impact on the design and construction of the railway and is the key area to prevent the occurrence of geothermal anomaly disasters. In some of the details, the difference in the prediction results is more significant. The high abnormal area obtained by the weights of entropy information model and the weights of evidence information volume model is larger, and the classification of the prediction results of the weights of evidence model is more significant, and we can clearly see the spatial correlation between the medium and low abnormal areas and high abnormal areas. The index-overlay model can obtain the smallest abnormal area, and the subtle abnormal changes cannot be effectively displayed in the classification map.

Additionally, there is a certain area of false anomaly in the eastern part of the study area, which is due to the impact of the urban heat source of Ya'an and Chengdu, causing the LST of the area rise. In the model with a large weight of LST, there will be a certain degree of interference in the judgment of geothermal anomalies.

## 4. Model Assessment

### 4.1. Analysis of Success Index

The occurrence rate of geothermal training points can be used to assess the effectiveness of the information model—that is, to analyze the success index of models. Compared with the prior probability of geothermal occurrence (0.00063%), it can indicate whether the prediction results are accurate [19]. The prior probability is the ratio of the total number of geothermal points to the total number of grids in the study area. According to the classification of the prediction results, a success index analysis table (Table 7) was obtained. It can be seen from the table that the occurrence rate of geothermal points will also increase with the increase in the anomaly degree of the geothermal anomaly area, which shows that the evaluation results of the information series model are reliable. In the prediction

of highly abnormal areas, the weights of the entropy information model do not reflect the advantages, and the success index is lower than that of the index-overlay model; the results of the weights of evidence information model in high and medium abnormal areas are better, indicating that the prediction results are more accurate.

**Table 7.** Success indices from the classification maps of prediction.

| Class | Index-Overlay | | | Weights of Entropy | | | Weights of Evidence | | |
|---|---|---|---|---|---|---|---|---|---|
| | Grids | Points | Index | Grids | Points | Index | Grids | Points | Index |
| 4 | 1,689,379 | 74 | 0.44% | 2,696,687 | 106 | 0.39% | 1,814,461 | 96 | 0.53% |
| 3 | 3,764,914 | 77 | 0.21% | 5,649,918 | 84 | 0.15% | 4,127,875 | 75 | 0.18% |
| 2 | 4,397,731 | 52 | 0.12% | 5,009,384 | 31 | 0.06% | 5,189,797 | 41 | 0.08% |
| 1 | 29,611,165 | 46 | 0.02% | 26,107,200 | 28 | 0.01% | 28,331,056 | 37 | 0.01% |

*4.2. Analysis of Area Ratio*

Area ratio analysis can be used to quantify the prediction results of the model. The effectiveness of the model was evaluated by calculating the area covered by the prediction probability function [57]. According to the attribute values of the prediction results of the information model, the results were arranged from large to small and divided into 25 classes by the equal quantile method. The number of geothermal points in each class and the grid area in the classification interval were assessed, the cumulative percentage curve of the grid units and points was calculated, then the prediction function curve was obtained. The expression of area ratio is as follows:

$$\lambda = (2A - P)/(2 - 2P), \tag{24}$$

where P is the prior probability (0.00063%); A is the area surrounded by the prediction probability function and X-axis; $\lambda$ is the area ratio of the prediction accuracy of the model. The closer the value is to 1, the more accurate the prediction result is. The calculation results are shown in Figure 8, and the area ratio of the models is 0.859, 0.846, and 0.872, respectively, which indicates that the overall accuracy of the model is higher, but the difference between these models is small. Additionally, the accuracy of the weights of evidence information model is slightly higher than that of the other two models.

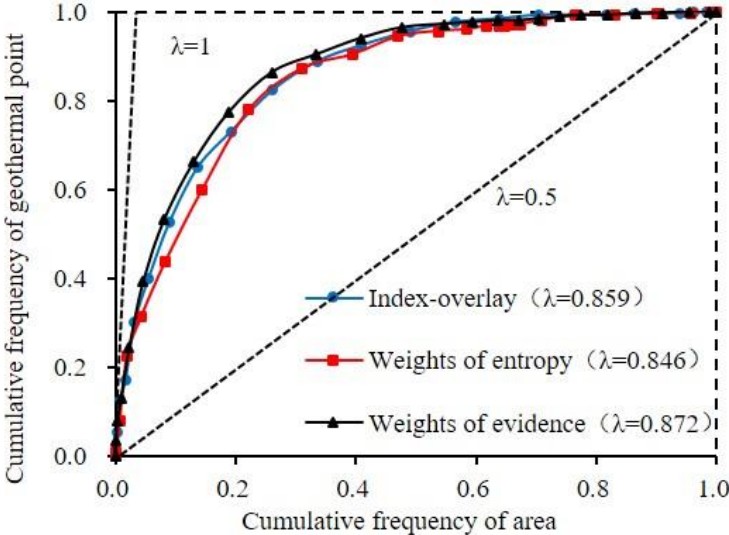

**Figure 8.** Curves of prediction ratio function and its area ratio.

*4.3. Ground Verification*

In order to further verify the effectiveness of the proposed model, we used twelve ground measurement points (Figure 9). Verification points were located in Guoqing,

Changdu City, Litang, Ganzi Prefecture and Bayi District, Linzhi City about 2100–8700 m away from the railway line and close to the river, with flat terrain, bare soil, and low vegetation; Points 1–6 belonged to the high geothermal anomaly area predicted by the weights of evidence model and points 7–12 were in the low geothermal anomaly area. The temperature of water and bedrock at a certain depth within 10 m$^2$ around the verification point is measured, and the specific situations of some verification points are shown in Table 8. It can be seen that the temperature range of the verification points in the high anomaly area is significantly higher than that in the low anomaly area. The ground test can verify that the prediction results of the weights of evidence information model are reliable and accurate to a certain extent.

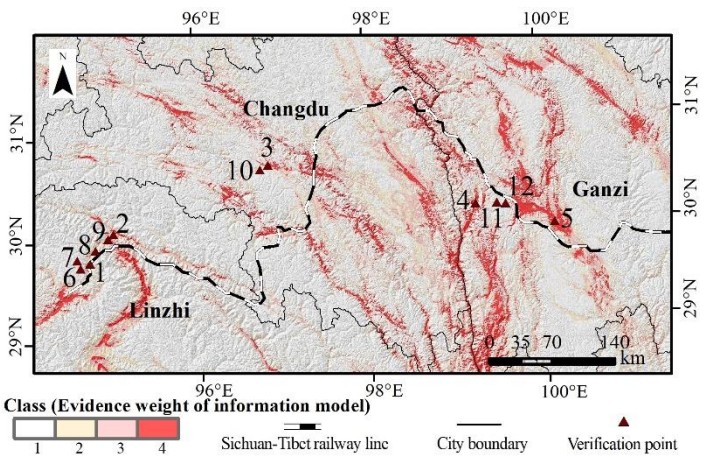

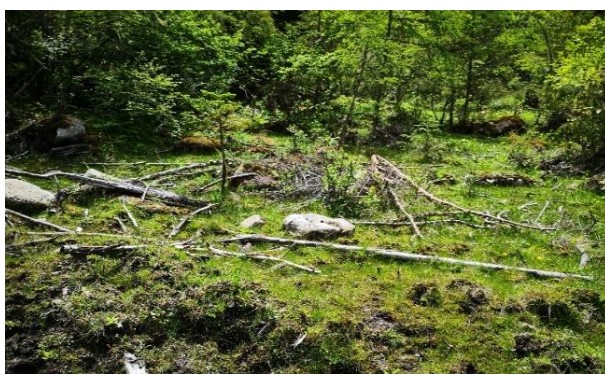

(**a**) Location of verification points      (**b**) Surrounding environment of verification point 6

**Figure 9.** Verification point location and surrounding environment. (**a**) Location of verification points, (**b**) Surrounding environment of verification point 6.

**Table 8.** Verification point details.

| Number | Measurement Date | Coordinate | Altitude (m) | Temperature (°C) |
|:---:|:---:|:---:|:---:|:---:|
| 1 | 12 June 2019 | 94°44′19″ E, 29°48′56″ N | 3175 | 30.8–39.7 |
| 2 | 12 June 2019 | 95°02′22″ E, 30°03′54″ N | 2033 | 50.3–64.6 |
| 5 | 9 June 2019 | 100°08′29″ E, 30°02′16″ N | 3971 | 31.4–52.3 |
| 10 | 12 June 2019 | 96°50′44″ E, 30°42′49″ N | 4377 | 19.3–33.2 |
| 11 | 9 June 2019 | 99°20′24″ E, 30°16′55″ N | 3297 | 17.9–23.9 |

## 5. Discussion and Conclusions

### 5.1. Discussion

High-temperature geothermal phenomena will seriously affect the construction of a plateau railway. In order to predict the area of geothermal high-temperature anomalies, a prediction method for the geothermal anomaly area was constructed based on the weighted information model for the Sichuan–Tibet railway area. Compared with the current prediction research on the geothermal anomaly area, we fully considered the geological factors, natural environment factors, and inducing factors of geothermal high-temperature anomalies and selected a number of impact factors closely related to the geothermal are, thus reducing the one-sidedness of single-factor or multifactor analysis.

Information models are widely used in geological hazard risk assessment [58], metallogenic prediction [59], hydrological feature extraction [60], and resource potential assessment [24]. At present, the superposition of factors is often used to accumulate the information of each factor to obtain the final evaluation results. However, the contribution of each factor to the prediction target cannot be the same, so the weighted superposition can comprehensively analyze the influence and relationship of multiple factors on the research target. The weight of evidence method has been extended to many fields, such as resource prediction [56], disaster analysis [61], and ecological environment research [62]. This weight determination method was introduced into the information model to enrich the research on the weight of information model, and it can quantitatively analyze the role of multiple factors. It can achieve better prediction results and greatly reduce the impact of human subjective factors on the prediction results. The model can provide geological information for the design and construction of the Sichuan–Tibet railway.

This paper optimizes the existing prediction methods used for geothermal anomaly areas, but there are still some problems that need to be fixed: on the one hand, the accuracy of the Bouguer gravity and magnetic anomaly data is relatively low, which will affect the prediction results to a certain extent; on the other hand, the prediction results of the geothermal anomaly areas are only classified and discussed in the macro scale. In the next stage of our research, we will make use of emerging geological remote sensing technologies such as reconstruction of Light Detection and Ranging (LiDAR), and the high and medium abnormal areas will be analyzed with a smaller spatial scale and in more detail in order to form specific guidance for the design and construction of the Sichuan–Tibet railway.

### 5.2. Conclusions

(1) In terms of impact factor selection, Landsat8 image data, various map data, and measured data were converted into LST, combined entropy of geological formation, buffer distance to faults, fault density, buffer distance to rivers, magnetic anomaly, Bouguer gravity, and earthquake peak acceleration data and combined with the measured geothermal points as the impact factors to predict geothermal high-temperature anomalies. After transformation, the spatial distribution of the factors had a certain relationship with the geothermal area: the more the geothermal points that were distributed in the areas with a higher LST, the higher the combination entropy was, the higher the fault density was, the closer it was to rivers and faults, the lower the magnetic values were, the higher the Bouguer gravity was, and the more severe the earthquakes were. In the factor correlation analysis, it was found that the correlation |R| of some factors was >0.3. Therefore, the buffer distance to faults and Bouguer gravity were removed, and the remaining six impact factors were retained as the input maps of the subsequent prediction model.

(2) Based on the information model, the objective weighted method was introduced into the prediction of the geothermal anomaly area and the index-overlay information model, the weights of entropy information model, and the weights of evidence information model are established, respectively. From the calculation results of the information values, different classes of reclassification factors were found to have different positive and negative effects on geothermal anomaly. Overall, with the increase in LST, combined entropy of geological formation, fault density, and earthquake peak acceleration, the values of the information provided were also increased. On the contrary, there was a negative correlation between the buffer distance to rivers and magnetic anomalies and geothermal high-temperature anomaly—with the increase in their values, the information values decreased. From the weight calculation results, the LST, fault density, and buffer distance to rivers accounted for a large proportion of the weights of entropy model, and the prediction results were mainly based on the information provided by these factors; the weight of LST, combined entropy, and fault density in the weights of evidence is relatively large. The results of the two models show that the core factors in judging geothermal high-temperature anomalies are LST and fault, which represent the geothermal flow and geological structure.

(3) The overall predictions of the three models were relatively similar. The geothermal anomaly area ran northwest–southeast, and the high anomaly area was mainly distributed in the middle of Ganzi and Changdu and the north of Linzhi. The fault structures and river systems in these areas are highly developed, the LST is high, and earthquakes occur frequently. Among them, the geothermal anomaly areas in Ganzi and Linzhi have a great impact on the design and construction of the railway. In the prediction details, the highly abnormal area obtained by the weights of entropy information model and the weights of evidence information model was larger, and the classification result of the latter is more prominent; the abnormal area obtained by the index-overlay model was the smallest, and the classification situation was relatively fuzzy in the map. From the analysis of the success index, the three models achieved a good prediction effect. The weights of evidence information model was the best in the prediction of the high abnormal area and the index reached 0.0053%, far exceeding the prior probability (0.00063%); the index-overlay model was better in the medium abnormal area and the index was 0.0021%. From the analysis of the area ratio, we can see that the prediction results of these models were similar, and the weights of evidence model has a slight advantage.

**Author Contributions:** W.Z. and Z.C.: methodology, writing-original draft preparation and editing, and software. T.F.: writing-review and editing. D.W.: validation and formal analysis. L.J.: in-vestigation and data curation. S.D.: resources and visualization. X.Z.: supervision and project administration. Q.D. and J.C.: funding acquisition. D.M. and M.B.: data arrangement and processing. All authors have read and agreed to the published ver-sion of the manuscript. All authors have read and agreed to the published version of the manuscript.

**Funding:** This research was funded by National Natural Science Foundation of China (No. 41876210), National Key Research and Development Program of China (No. 2017YFA0603003 and No. 2017YFC0 601502) and Institute International Cooperation Projects (E03405020N).

**Institutional Review Board Statement:** Not applicable.

**Informed Consent Statement:** Not applicable.

**Data Availability Statement:** Not applicable.

**Acknowledgments:** We would like to offer special thanks to NASA and USGS for providing us with the Landsat8 images. We also thank the China Railway Eryuan Engineering Group Co. Ltd. for the support of field investigation and geothermal point data they provided. A special acknowledgement should be expressed to China-Pakistan Joint Research Center on Earth Sciences that supported the implementation of this study.

**Conflicts of Interest:** The authors declare no conflict of interest.

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
