# Peer review of "Weighted Information Models for the Quantitative Prediction and Evaluation of the Geothermal Anomaly Area in the Plateau: A Case Study of the Sichuan–Tibet Railway"

_remotesensing, doi:10.3390/rs13091606_

Round 1

Reviewer 1 Report

    I read the revised version of your manuscript. I find it interesting, and I think that it has the potential to be published. However, it is my perception that several aspects can be clarified, and especially the English expression. At present, the text is muddled and confused, affecting the clarity of the story you want to tell. All the sections require a deep review. It is your responsibility to clean up the entire manuscript and submit a better version of your work. Send the text to a native English-speaker for review.
    Another question is related to figures quality. It would help if you had more attention to the font size of them. Please, consider removing some figures.

Author Response

Dear Editors and Reviewers:

Thank you very much for your constructive and positive comments on our manuscript entitled “Weighted Information Models for the Quantitative Prediction and Evaluation of the Geothermal Anomaly Area in the Plateau: A Case Study of the Sichuan–Tibet Railway” submitted to Remote Sensing.

We have revised our paper along the lines outlined by the reviewers. Our detailed responses follow, and the reviewers’ comments are in bold. Note also that changes in the manuscript are in yellow colour.

Response to Reviewer 1 Comments

Point 1: I read the revised version of your manuscript. I find it interesting, and I think that it has the potential to be published. However, it is my perception that several aspects can be clarified, and especially the English expression. At present, the text is muddled and confused, affecting the clarity of the story you want to tell. All the sections require a deep review. It is your responsibility to clean up the entire manuscript and submit a better version of your work. Send the text to a native English-speaker for review.

Reply: We have sent our manuscript to MDPI for English editing (English editing ID: 28230) and the English revision has been completed. We think the text in this manuscript has become clearer and easier to understand.

Point 2: Another question is related to figures quality. It would help if you had more attention to the font size of them. Please, consider removing some figures.

Reply: We further enlarge the size of the figures so that the readers could see the labels and legends on the figures better. Due to the limitation of figure size, we can't show more detail information on the figure, so the original figure 7 (the model prediction results without reclassification) seems to have basically the same pattern, so we delete the original figure 7 directly and only retain the prediction results after reclassification.

Special thanks to you for your positive and constructive comments for our work.

Reviewer 2 Report

Thanks to the authors for modification, the manuscript quality quite better and insert some new map layers and more details in the methodology. I have some minor comments such as abstract keywords not suitable, font of map legend are very small, the section 5.2 about a discussion is better to come before conclusion either write at the end of result section, check the English grammatical errors and references.

Author Response

Thank you very much for your constructive and positive comments on our manuscript entitled “Weighted Information Models for the Quantitative Prediction and Evaluation of the Geothermal Anomaly Area in the Plateau: A Case Study of the Sichuan–Tibet Railway” submitted to Remote Sensing.

We have revised our paper along the lines outlined by the reviewers. Our detailed responses follow, and the reviewers’ comments are in bold. Note also that changes in the manuscript are in yellow colour.

Response to Reviewer 2 Comments

Point 1: Thanks to the authors for modification, the manuscript quality quite better and insert some new map layers and more details in the methodology. I have some minor comments such as abstract keywords not suitable, font of map legend are very small, the section 5.2 about a discussion is better to come before conclusion either write at the end of result section, check the English grammatical errors and references.

Reply: We re-extract the keywords of this manuscript and further enlarge the size of the figures so that the readers could see the labels and legends on the figures better. And we put section 5.2 about discussion before conclusions. In terms of English revision, we have sent our manuscript to MDPI for English editing (English editing ID: 28230) and the English revision has been completed. We think the text in this manuscript has become clearer and easier to understand.

Special thanks to you for your positive and constructive comments for our work.

Reviewer 3 Report

Minor english review is recommended.

Follow comments on annotated pdf file for formal changes in graphs and tables. 

Author Response

Thank you very much for your constructive and positive comments on our manuscript entitled “Weighted Information Models for the Quantitative Prediction and Evaluation of the Geothermal Anomaly Area in the Plateau: A Case Study of the Sichuan–Tibet Railway” submitted to Remote Sensing.

We have revised our paper along the lines outlined by the reviewers. Our detailed responses follow, and the reviewers’ comments are in bold. Note also that changes in the manuscript are in yellow colour.

Response to Reviewer 3 Comments

Point 1: Minor English review is recommended.

Reply: We have sent our manuscript to MDPI for English editing (English editing ID: 28230) and the English revision has been completed. We think the text in this manuscript has become clearer and easier to understand.

Point 2: Follow comments on annotated pdf file for formal changes in graphs and tables.

Reply: We revise the manuscript according to the comments in the pdf. Due to the limitation of figure size, we can't show more detail information on the figure, so the original figure 7 (the model prediction results without reclassification) seems to have basically the same pattern, so we delete the original figure 7 directly and only retain the prediction results after reclassification. And in the part of ground verification experiment, we mark all 12 verification points on the figure 8, and select 5 of them for further explanation, in order to show that the design and implementation of our verification experiment are more scientific, which can verify the effectiveness of the proposed model.

Special thanks to you for your positive and constructive comments for our work.

Round 2

Reviewer 1 Report

Please, revise the figures with tiny fonts.